# Equity or Equality? Which Approach Brings More Satisfaction in a Kidney-Exchange Chain?

**DOI:** 10.3390/jpm11121383

**Published:** 2021-12-18

**Authors:** Arian Hosseinzadeh, Mehdi Najafi, Wisit Cheungpasitporn, Charat Thongprayoon, Mahdi Fathi

**Affiliations:** 1Industrial Engineering Department, Sharif University of Technology, Tehran 1458889694, Iran; hosseinzade.arian@ie.sharif.edu (A.H.); najafi.mehdi@sharif.edu (M.N.); 2Ted Rogers School of Management, Ryerson University, Toronto, ON M5G 2C5, Canada; 3Department of Medicine, Mayo Clinic, Rochester, MN 55905, USA; charat.thongprayoon@gmail.com; 4Department of Information Technology and Decision Sciences, G. Brint Ryan College of Business, University of North Texas, Denton, TX 76201, USA

**Keywords:** kidney exchange, kidney chain donation, equity, graft survival prediction, financial neutrality

## Abstract

In United States (U.S.), government-funded organizations, such as NLDAC, reimburse travel and subsistence expenses incurred during living-organ donation process. However, in Iran, there is a non-governmental organization called Iranian Kidney Foundation (IKF) that funds the direct and indirect costs of donors through charitable donations and contributions from participants in the exchange program. In this article, for countries outside the U.S. that currently use an equality approach, we propose a potential new compensation-apportionment approach (equitable approach) for kidney-exchange chains and compare it with the currently available system (equality approach) in terms of the apportionment of compensation in a kidney-exchange chain to cover the expenses incurred by the initiating living donor of the chain in the act of donation. To this end, we propose a mechanism to apportion compensation among all participating pairs based on the equity approach by utilizing a prediction model to calculate the probability of graft survival in each transplant operation. These probabilities are then used to define the utility of any transplantation, considering the quality of each pair’s donated and received kidney in the chain. Afterward, the corresponding cost is apportioned by a mechanism based on the normalized differences between the utility of donated and received kidneys for each incompatible pair of the chain. In summary, we demonstrate that by utilizing the equitable approach, there is more fairness and equity in the allocation of resources in organ-procurement systems, which results in more satisfaction among incompatible pairs. Additional future prospective studies are needed to assess this proposed equitable approach for kidney-exchange chains in countries outside the U.S., such as Iran, that currently use an equality approach.

## 1. Introduction

The two currently available treatment options for patients with kidney failure are dialysis or kidney transplantation [1]. Although both treatments are used for end-stage kidney disease (ESKD) patients, kidney transplantation results in better survival and quality of life (QOL) [2,3,4]. Therefore, it is not surprising that kidney transplantation is the preferred treatment for ESKD patients. Kidney allografts used for transplant operations can come from deceased donors or living donors [5].

Patients with an incompatible living donor can participate in kidney-exchange programs. Kidney exchange or kidney paired donation (KPD) involves the exchange of kidneys between living donors and recipients so as to facilitate transplantation of incompatible pairs and optimize available resource utilization. Blood group O recipients are often disadvantaged in this process [6,7]. KPD can occur between two incompatible pairs (two-way swap) or through a chain whereby a chain starter, often a non-designated donor, who is a member of blood group O, donates altruistically. This kidney is used to initiate a chain of kidney exchanges between incompatible donor-recipient pairs [8]. This dependency on a living non-designated donor intuitively blunts the transplantation process. In other words, according to the operational aspects of kidney-exchange chains, all patients in the chain have to wait until the authorized entity finds an altruistic donor to initiate the chain.

There are financial barriers for non-designated living donors, despite their altruism. Living donors face both direct and indirect costs for their act of donation. Direct costs are related to transportation, lodging and parking, whereas indirect costs include lost wages, use of paid time off, dependent care expenses and risks to job stability [9,10,11,12,13,14]. Eliminating these financial disincentives may result in an increase in access to living-donor transplantation [9,11,12]. To eliminate these factors, different measures have been taken in different countries. For instance, in United States (U.S.), recipients are allowed to pay for a donor’s travel, housing and lost wages, according to the National Organ Transplant Act of 1984 (NOTA), although this forbids organ purchases [9,14]. To this end, the National Living Donor Assistance Center (NLDAC) was established in 2007 to reimburse travel and subsistence expenses incurred toward living organ donation. Thenceforth, NLDAC has been funded through a federal grant and has supported applicants with significant financial hardship [9,13]. Furthermore, other means have been set up to raise money to cover the costs of living donors. These efforts include bake sales, fundraising events, crowdfunding, etc., [14]. There have been other measures adopted in other countries outside the U.S. For example, in Iran, there is no government-funded program developed to cover the direct and indirect costs associated with donation. However, an organization called the Iranian Kidney Foundation (IKF) has been established that funds donors’ direct and indirect costs through charitable donations and contributions from participants in the exchange program [15,16,17].

Given to the current process of the IKF, the amount of money that should be paid from participants varies under different circumstances. To determine these amounts, the IKF tries to fund a part of compensation, usually about 30%, from the individual donors or organizations once the kidney-exchange chain and the initial donor are identified [18]. The remaining compensation is funded equally by the participants in the chain [15,19]. The equal apportionment of compensation is dissatisfactory for some participants who have better conditions in terms of age, lifestyle and health situation that affect the quality of kidney each pair donates and receives [20]. To overcome this drawback in countries outside U.S. that currently utilize the equality approach, we aim to develop a cost-apportionment mechanism based on equity and to compare its results with the current approach, which based on equality. In other words, we analyze two following approaches:

Equality approach: equality is easily reached by allocating an equal amount of the living donor’s costs to all pairs participating in the exchange plan. We named this approach the equal compensation amount (ECA) approach.

Equitable approach: unlike the ECA approach, this approach is complicated and aims to consider equity in the apportionment of the non-designated donor’s costs among the incompatible pairs in the exchange plan. Regarding the focus of this approach on fairness, it is named the fair compensation amount (FCA) approach.

Regarding these definitions, we aim to develop a mechanism to adopt the FCA approach in compensation apportionment. We also compare the results achieved by this approach with those achieved by the ECA to determine which approach brings more satisfaction to participants in the chain. Note that the developed FCA approach can also be employed in transplantation systems in which government-funded programs have limited funding and cannot cover all expenses of living donors.

The rest of the article is organized as follows: In Section 2, a review of the related literature is presented. Section 3 defines the problem under investigation, presents the assumptions and methodology used in the current study and develops a mechanism to apportion compensation among patients regarding the FCA approach. Section 4 presents numerical examples, investigates the operational characteristics of the proposed mechanism and compares the apportionment results of the FCA and ECA approaches. Finally, conclusions and several recommendations for future research are presented in Section 5.

## 2. Literature Review

Kidney exchange between incompatible pairs was first proposed in 1986 by Rapaport [21]. The simplest method of KPD is a two-way exchange. This method of kidney exchange was first implemented in 1991 in Korea [22]. Since then, owing to operational innovations, models used in KPD have become considerably more advanced. Three-way and four-way exchanges are examples of other implemented exchange models between incompatible or compatible pairs [23,24].

As mentioned earlier, chain kidney donation is another approach in kidney-exchange programs, which has been widely discussed in living donor availability [8,25,26,27,28,29]. Due to the variations in types of kidney exchange methods and different circumstances in operational exchange programs worldwide, operations research has been utilized to maximize the benefits of these methods [30].

Prediction of future events has been a matter of attention for decades; however, prognostic studies only began to receive adequate attention a few years ago [31]. Prediction models in medicine, especially in the last decade, have been designed to help physicians predict the risk of events in patient-related decision-making processes [32]. The ability to predict graft survival among kidney-transplant recipients is critical in allocating donors to patients since the rate of success in finding compatible donors is always limited. Therefore, many researchers have worked on prediction models for kidney-graft survival, classified into three categories: simulation and operation research, conventional statistics, and data analytic approaches [33]. A known subcategory of conventional statistical studies is the Cox proportional hazards model, a widely used multivariate approach in medical literature to assess survival time, which can be utilized for categorical and numerical types of predictors [34,35,36,37,38]. This model aims to evaluate the effects of several variables/covariates on the rate of a specific event (e.g., graft failure) at a particular point in time.

Regarding the mentioned limitations of kidney-exchange programs, the main contributions of this work are summarized below:

We aim to (1) propose a new equity-based approach to achieve financial neutrality of the initiating non-designated donor in kidney-exchange chains in sharing the direct and indirect costs of the living donor between the participating incompatible pairs that can be considered outside the U.S., and (2) compare two different approaches, ECA and FCA, in terms of apportioning the required compensation among chain participants.

To implement the FCA approach for the apportionment of the participant compensation, we must take the characteristics of all donated and received kidneys in the chain into account. To address this issue, we develop a mathematical mechanism that determines each pair’s share regarding the attributes its donated and received kidneys.

The proposed mechanism determines the probability of graft survival by utilizing multivariate Cox regression survival analysis [20,39], converts it to transplant’s utility and determines the compensation share according to the obtained extra utility as a more equitable attribute for compensation sharing.

## 3. Problem Definition and Formulation

To define and formulate the problem under investigation, we first describe the problem and list its assumptions. Then, we discuss the methodology and approaches used to apportion compensation among the available pairs in a kidney-exchange chain.

### 3.1. Problem Definition and Underlying Assumptions

As mentioned in the preceding section, the purpose of this article is to compare ECA and FCA approaches in the operation of apportioning the compensation of external donors who starts a kidney-exchange chain among the participating incompatible pairs and to present a mathematical mechanism for this goal under the FCA approach. The share of the total compensation allocated to each incompatible pair in the chain under the FCA approach must be according to the characteristics to both kidneys being receives and donated to the chain.

To describe the problem, Figure 1 illustrates a kidney-exchange chain with a non-designated living donor, four incompatible pairs and a recipient from the waiting list. Intuitively, several recipients and donors in the chain have different characteristics, such as age, sex, blood type, and body size. Therefore, they need to find a donor whose kidney is compatible with the patient of pair 1. Though all incompatible pairs pay the compensation altogether, they may not have the same payment share due to their variant situation.

Several authors in the literature have discussed the notion of fairness and equity in the allocation of divisible and indivisible goods [40,41,42,43,44]. In general, an allocation is defined as equitable if no agent envies another, i.e., no agent prefers another agent’s bundle to his own. Additionally, an allocation is defined as fair if it is both equitable and Pareto efficient. Therefore, to have a fair allocation, one must minimize envy between agents. In our problem, agents are incompatible pairs in the chain, and bundles are the recipients’ allocated kidneys. Therefore, it is possible that an incompatible pair envies another incompatible pair’s place in the chain if the exchange of places is feasible between those two agents. However, since an optimization model already forms the exchange chain, any exchange of places between its participating pairs will result in less overall transplant quality and, consequently, more overall envy. Therefore, the allocation of places in each chain between participating pairs has the minimum feasible overall envy.

The main question of this article is how this compensation should be apportioned among incompatible pairs to have the minimum overall envy after the operations. To answer this question, this study presents a mathematical mechanism to determine the share of compensation paid by each participating pair under the FCA approach. Then, the results from this mechanism are compared with the apportionment results under the ECA approach. The following major assumptions are considered:

The exchange chain is given in this study. To obtain this chain, one can employ approaches available in the literature [5,45,46]. Therefore, the characteristics of incompatible pairs are known, and there is no limitation for the length of the chain used in the proposed approach.

Living donor’s cost apportionment is carried out to determine the share of each participating pair before the transplantation process.

The chain pairs have been chosen from a predefined pool and do not choose to move to another chain [8]. That is because of two facts: first, they are not aware of the existence of any other chain; second, they are not aware whether the other chain, if available, is more appropriate for them or not.

All donor-recipient participants in the chain program are incompatible, and their orders in the chain have been defined based on the exchange program’s objective. Therefore, an incompatible pair cannot alter its corresponding pair for the kidney exchange [8,25].

It is assumed that all of the transplants in the chain would be performed successfully. In case of failure of the exchange chain due to last-minute failure of a transplant, the total share of subsequent incompatible pairs from the compensation is paid by the organization responsible for organ procurement and transplantation in the system.

The characteristics of the kidney initiating the chain program are known. The amount of compensation that should be paid to the initial donor to cover his costs associated with the donation process is also given, and all participants are informed about and have agreed to this value.

Although the last donor-recipient pair of the chain pays a share of the corresponding patient’s compensation, their donated kidney would be assigned to the first compatible patient on the waiting list.

### 3.2. Methodology

We compared two different approaches based on equality and equity to determine the share of the compensation for incompatible pairs. If *C* is the total amount of compensation, *n* is the number of participating incompatible pairs in the chain and *C_i_* is the portion of compensation allocated to incompatible pair *i*. In the ECA approach, *C_i_* is easily calculated as follows:(1)$$C_{i}=\frac{C}{n}\;\forall i\in \;\left\{1.2,\ldots ,n\right\}$$

However, the FCA approach is complicated. To develop this approach, we employ the Shapley value, one of the most prominent ways to allocate gains obtained by a set of players in coalitional cooperative games. The main idea of Shapley value is that members should receive payments or shares proportional to their marginal contributions in a cooperative game [47]. Allocation of costs in a cooperative game is another application of the Shapley value [48,49,50,51]. To employ the Shapley value for the purpose of compensation apportionment in the problem under investigation, the expected marginal contributions of each incompatible pair must be calculated [47]. In other words, compensation apportioning should match the characteristics of the kidney that participants are receiving from the system and the kidney they are donating to the system. For instance, receiving a younger kidney should result in a higher share, or conversely, donating a younger kidney should reduce the pair’s share.

Taking into account all mentioned factors, we calculate the odds of graft survival and determine patients’ share in providing compensation according to these probabilities. To this end, we employ the utility function to capture the nonlinear impact of graft-survival probability on the satisfaction of the pairs. Afterwards, compensation is apportioned among all pairs of the chain, considering their utility to the system. Given *U_i_* is the utility that transplant *i* brings about the system, the FCA approach is defined as:(2)$$C_{i}=f\left(C,u_{i-1},u_{i}\right)\;\forall \;i\;\in \;\left\{1.2,\ldots ,n\right\}$$
where *u_i_* is the utility donated by the *i*th incompatible pair to the chain and equivalents to the utility of the *i*th transplant, estimated based on the system administrator’s preferences as the entity responsible for matching donors and patients. In addition, *u_i−_*_1_ indicates the amount of utility received by incompatible pair *i* and, similarly, equals the utility of transplantation *i*−1. Accordingly, *u*_0_ indicates the utility of the initiating transplantation of the chain.

Note that each transplant has a utility value for the social planner, e.g., the IKF, aiming to maximize the number of patients receiving a kidney in the exchange program. Since Equation (2) takes the condition of both received and donated kidneys into account, it diminishes total envy between participants and consequently increases fairness.

Utility theory has been proposed by Von Neumann and Morgenstern [52] to measure a decision maker’s preferences in a mathematical form. To apply this theory, a function known as a utility function can generally be estimated to calculate a decision maker’s preferences between various choices in a decision-making problem [53]. According to this theory, if the decision maker prefers choice A to other choices, the utility of choice A must be higher than that of other choices [52]. In addition, another characteristic of the corresponding utility function is monotonicity. A monotonically increasing utility function means that the decision maker prefers higher values of an attribute.

Conversely, the decision maker prefers lower values of an attribute if the utility function is monotonically decreasing. With a non-monotonic utility function, the decision maker’s preferences concerning different attribute levels may increase or decrease [53]. Risk aversion is a characteristic of the decision maker that can be inferred from the utility function. In general, a decision maker is risk-averse if and only if his/her utility function is concave. On the contrary, risk-prone decision makers have convex utility functions [53]. Utility theory has been widely used in healthcare problems, such as measuring patients’ quality of life based on their condition [54,55,56] and measuring quality of life for ESKD patients on dialysis [57].

To estimate the utility of a transplant, different characteristics of the recipient and the donor must be considered. Therefore, rather than directly estimating the utility function based on these attributes, we propose using these attributes to estimate the probability of graft survival and then defining each transplant’s utility based on its graft-survival probability. In this article, the likelihood of graft survival is estimated by a calculator based on the multivariate Cox regression survival analysis presented in [20]. This calculator, called the kidney graft survival calculator (KGSC), developed based on 15 years of United States transplantation data from the Scientific Registry of Transplant Recipients (SRTR), can estimate the probability of graft survival 5 and 10 years post-transplant. It takes into account several attributes of donors and recipients, such as age, sex and body size. It also considers the type of the living donor (related/unrelated), number of HLA ABDR mismatches, ABO compatibility between donor and recipient, donor and recipient race, donor’s history of cigarette use, transplant year, recipient’s panel reactive antibody (PRA), time on dialysis, insurance type, history of previous transplants and recipient’s diabetes status the potential impact of these factors on the probability of graft survival. Regarding the variant of attributes utilized in this calculator and its ability to estimate the likelihood of graft survival for transplants with living donors [20], we chose KGSC in our proposed equitable-cost-apportionment mechanism. Additionally, we established our mechanism based on the probability of graft survival 10 years after transplant, since this period is frequently considered and used in graft-survival predictions and evaluations in the literature [20,58,59,60,61].

As mentioned, converting the graft-survival probability to the transplant’s utility requires definition of a utility function. This study estimated this function using the standard gamble (SG) technique [62] on the data gathered from 10 selected experienced experts of the Iranian Kidney Foundation who are in charge of the matching process in Tehran, Iran. To utilize this technique, we designed an interview structure and asked these experts to state their preferences of different probability levels of graft survival. According to this structure, presented in Figure 2, each interviewee was subjected to seven sets of lotteries in three stages. In each lottery, based on the SG technique, the participant was asked to imagine a situation where they had two alternatives to assign an unknown kidney to a recipient. The first alternative was participating in a gamble in which the graft-survival probability after transplant was one of two known values with a 50-50 chance. The second alternative was undergoing the transplant with a different fixed value of graft-survival probability. After knowing the graft-survival probability values of the two probable outcomes of the first alternative in each lottery, the respondents were asked to declare the minimum probability of graft survival they would still prefer as the second alternative rather than the first alternative. The utility of the declared value is the average of utilities of the outcomes of the first alternative. As presented in the Appendix A, for simplicity, the participants were asked to declare this value as a number multiple of 5%. After calculating the utility of the declared value of survival probability, by replacing the gamble outcomes one by one with this value, two different lotteries of the next stage were designed. After receiving each of the interviewees’ responses to all seven lotteries, the average probabilities of graft survival in each level of utility were estimated. It is worth mentioning that since the utility values are defined in the interval between 0 and 1, we could not increase the number of the independent values extracted from the participants, so calculation of utility function was based on these values.

### 3.3. Mechanisms of Compensation Apportionment

As Equation (1) shows, the ECA approach apportions the identified compensation among all participants equally. However, the mechanism proposed under the FCA approach aims to apportion compensation among all participants regarding the characteristics of both kidneys they will receive and donate. These characteristics can be represented by the amount of utility provided for each participant. Due to the impacts of transplantation failure on the recipient’s quality of life and length of the waiting list, a kidney’s survival probability in the recipient’s body is usually considered the main factor in patient-utility estimation [63]. According to this factor and based on our interview structure, presented in Section 3.2, the utility values of transplants and the corresponding values of graft-survival probability were obtained from the interviewees. Utility values of 0.0% and 100.0% survival probability were assumed at 0.0 and 1.0, respectively, to initiate the interview process. The results of these interviews are presented in the Appendix A. In addition, the average results of the interviews are shown in Table 1. Based on these results, Figure 3 illustrates the relationship between the probability of graft survival and transplant utility. As this figure shows, the obtained utility function is monotonically increasing and concave, demonstrating that decision makers are risk-averse [53].

Regarding the collected data, provided in the Appendix A and presented in Table 1 and Figure 3, and the characteristics of the utility function being appropriate to capture this behavior [52], transplant’s utility function is estimated as follow:(3)$$U=1.32077-1.32847\times \exp\left(-1.46741\times T\right)$$
where *U* is the estimated utility of the transplant and *T* is the probability of graft survival 10 years after transplantation.

Although the acquired utility function helps us estimate any transplantation utility, it could not be directly utilized to determine the share of an incompatible pair. Each pair grants an amount of utility to the chain and receives another amount from the chain. Therefore, the difference between the granted and the received utilities should be considered as an index to determine the share of compensation. To this end, let us assume that *n* is the number of incompatible pairs, $\Delta u_{i}$ is the difference between the granted and the received utilities for incompatible pair *i*, that is the net utility, and be calculated as follows:(4)$$\Delta u_{i}=u_{i-1}-u_{i}\;\forall i\in \;\left\{1.2,\ldots ,n\right\}$$

To avoid a negative value of Δ*u* and to keep the share of incompatible pairs non-negative, Equation (4) is normalized as follows:(5)$$\Delta u_{i}^{\prime }=\left\{\begin{array}{cc} \Delta u_{i}+\left|\underset{j}{min}\Delta u_{j}\right| & \forall j\in \;\left\{1.2,\ldots ,n\right\}\;if\;\underset{j}{min}\Delta u_{j}<0\\ \Delta u_{i} & \forall j\in \;\left\{1.2,\ldots ,n\right\}\;if\;\underset{j}{min}\Delta u_{j}\geq 0 \end{array}\right.$$

In this Equation $\Delta u_{i}^{\prime }$ is the normalized value of the net utility of incompatible pair *i*. Regarding Equation (5), an incompatible pair pays more portion of compensation if it has a higher value of normalized net utility. Therefore, each pair’s percentage of total compensation is calculated as follows:(6)$$C_{i}=\left\{\begin{array}{cc} \frac{C}{n} & if\;\underset{i}{\sum }\Delta u_{i}^{\prime }=0\\ C*\frac{\Delta u_{i}^{\prime }}{\sum_{i}\Delta u_{i}^{\prime }} & if\;\underset{i}{\sum }\Delta u_{i}^{\prime }>0 \end{array}\right.\;\forall i\in \;\left\{1.2,\ldots ,n\right\}$$
where *C_i_* is the portion of the compensation allocated to incompatible pair *i* and *C* is the total amount of compensation that should be paid to the external donor. It is worth mentioning that if all pairs receive the same utility, the total normalized net utility would be zero. In this case, all pairs should pay an equal portion of the total compensation. Conversely, if the pairs receive variant normalized net utility, they would pay a different portion of the total compensation that could be obtained by Equation (6). Additionally, according to this Equation, the incompatible pair with the lowest Δ*u* value does not pay any share of the compensation if its respective Δu≤0.

## 4. Comparing Equity and Equality Approaches

To compare the results of cost apportionment under ECA and FCA approaches, first, we must evaluate the validity of our proposed cost-apportionment mechanism under the FCA approach. Therefore, Section 4.1 includes investigations of two different scenarios as a numerical study to this end. Then, in Section 4.2, we implement the comparison by utilizing a simulation model.

### 4.1. Numerical Study

To analyze the properties of the developed mechanism, we investigate two different scenarios of an operational kidney-exchange chain and discuss how the chain’s attribution mechanism and the amount of total compensation affect the compensation portion each participating incompatible pair should pay. The first scenario assumes four incompatible pairs (i.e., five kidney transplantations) with different characteristics available in the kidney-exchange chain. A woman with a high BMI (body mass index) in the age range of 50–59 donates her kidney to the exchange chain. Besides, the total compensation for this donation is considered $5000.

The latter scenario also includes four incompatible pairs similar to those in the former scenario. However, the external donor is a man with a healthy BMI and in the age range of <30. In addition, the living donor’s total cost in this scenario is set to $5500. Table 2 presents all required characteristics of transplants for estimating the probability of graft survival. As previously mentioned, the donor of transplant 1 is the external donor of the chain, and the recipient of transplant 5 is a patient from the waiting list. Additionally, the structure of the data in Table 2 is similar to that presented in [20].

Considering the data presented in Table 2 and the proposed mechanism, the probability of graft survival after 10 years and the corresponding utility of each transplantation are calculated and shown in Table 3. As shown in Table 3, the better characteristics of the external donor in scenario 2 lead to an increase of 6.3% in the probability of graft survival and 0.06 in the utility of the first transplantation. Since the amount of utility for the remaining transplantations is unchanged, the share of the total compensation assigned to the first incompatible pair in scenario 2 is expected to be higher than in scenario 1. Table 4 and Table 5, respectively, present the share of each incompatible pair in scenarios 1 and 2.

As the results show, incompatible pair 2 is not supposed to pay any portion of the compensation since it has the minimum negative net utility in both scenarios. In addition, a significant part of the compensation is assigned to incompatible pair 3 due to the positive value of its net utility. Furthermore, the obtained result shows that finding a non-designated living donor who increases the graft-survival probability in the first transplantation certainly increases the share of the first incompatible pair and decreases the share of other participants, except for those pairs whose shares are zero.

### 4.2. Numerical Analysis

In this section, intending to assess the functionality of the mechanism under the FCA approach and compare its results with the apportionment scenario under the ECA approach, we implement 1000 rounds of simulations of kidney-exchange chains. Through this process, each pair’s share of the compensation and the outcome of each transplant operation 10 years after the procedure is determined. All of the characteristics of transplants in the chains are generated using uniform distributions within their parameter ranges specified by the KGSC [20]. We perform a total of 1000 simulations, each representing a chain that contains 6–10 incompatible pairs. Additionally, the amount of compensation is set to $5000 for all of the chains. Figure 4 represents the frequency distribution of different chain lengths in our simulation.

Accordingly, these 1000 kidney-exchange chains include 8920 transplants in total, with random characteristics. According to our mechanism based on the FCA approach, the 10-year probability of graft survival and the utility value can be estimated for each of these transplants; Figure 5 and Figure 6 illustrate the distribution of these values for our total number of transplants, respectively.

Furthermore, Table 6 shows the frequency of each possible outcome for 8920 transplants derived from the simulation process. According to this table, 55.7% of transplants in our analysis did not survive for 10 years after transplantation.

The final results of the simulation process are presented in Table 7. As shown in this table, cost apportionment is implemented based on the FCA and ECA approaches. The former utilizes our proposed cost-apportionment mechanism to determine the share of each participant of the total compensation. The latter determines the share of participants by dividing the compensation equally among them. Additionally, there are two possible outcomes for each transplant after 10 years: graft survival or graft failure. Therefore, there are two possible outcomes for the received transplants of each incompatible pair. Table 7 includes the cost-apportionment results derived from the simulation process for these two possible outcomes by utilizing two approaches.

As previously mentioned, we aim to compare the results of cost apportionment based on two different approaches, ECA and FCA. In this regard, the functionality of these approaches must be assessed in the simulation results. As in Table 7, the total share of compensation for participants who received a kidney that survived 10 years after the transplant is determined to be 52% of the total compensation via the FCA approach and 44% via the ECA approach. Additionally, the average compensation contribution for these pairs is defined as $740 via the FCA approach and $629 via the ECA approach. In the results of the FCA approach, incompatible pairs that received successful transplants pay a higher amount of money than the according to the results of the ECA approach. On the other hand, the total share of compensation for incompatible pairs that received a kidney that did not survive is determined to be 48% of the total compensation via the FCA approach and 56% via the ECA approach. Additionally, the average compensation contribution for these pairs is determined to $545 via the FCA approach and $633 via the ECA approach. Thus, in the results obtained from the FCA approach, incompatible pairs that received unsuccessful transplants pay a lower amount of money than according to the results of the ECA approach. In addition, 767 of these incompatible pairs did not pay any amount of money by using the cost-apportionment mechanism based on the FCA approach, compared to the other outcome, in which only 233 participants did not pay. Generally, by analyzing the results obtained from the ECA and FCA approaches, it can be inferred that our proposed cost-apportionment mechanism seems to function in alignment with the goal of the FCA approach, which is moving towards more fairness and equity in the allocation of costs and resources in organ-procurement systems.

## 5. Conclusions and Future Research

In current renal-transplantation systems, living donors face financial burdens related to their act of donation, which can act as disincentives to donate. These financial burdens include transportation costs, lodging and parking costs, lost wages, etc. In many transplantation systems in other countries outside the U.S., such as Iran, organizations and governmental funds cannot cover these expenses. In these situations, compensation of the initiating living donors of kidney-exchange chains may potentially be paid by the participating pairs in the chains. We proposed the apportionment of the compensation of the initiating living donor in kidney-exchange chains between incompatible pairs of the chain based on the notion of equity, and compared its final results to those derived from the equality approach.

To achieve this goal, this study developed a new mechanism employing the kidney chain of exchanges and utility function to apportion compensation among participating incompatible pairs based on the FCA approach. The proposed mechanism considers all donor and recipient characteristics in the chain and attempts to determine a compensation ratio according to the utility value of pairs. According to this mechanism, the amount of compensation allocated to incompatible pairs increases as they receive a better-matched kidney, and their share decreases as they donate a better-matched kidney.

Furthermore, in comparison with dividing compensation equally between all participating pairs according to the ECA approach, the apportionment of total compensation between incompatible pairs by the proposed mechanism based on the FCA approach results in a decrease in total envy between the agents and an increase in fairness of the apportionment model. The reason for this is that by utilizing our mechanism, the apportioned share of compensation to agents with better bundles will be higher than that of other agents. Therefore, the amount of envy in their bundles will be lower. Thus, it can be concluded that by utilizing the FCA approach, we are moving towards more fairness and equity in the allocation of resources in organ-procurement systems, which results in more satisfaction among incompatible pairs.

It is worth mentioning that in the proposed mechanism, each transplant’s utility is calculated solely based on the probability of graft survival post-transplantation. It does not investigate other factors, such as patient health status, affordability, etc. In addition, this FCA approach is not developed for the U.S.; instead, it is mainly intended for other countries that currently use an equality approach, such as Iran, and we do not endorse the use of this FCA approach in the U.S.

The following are suggestions of topics that future research could investigate to improve the concept we presented in this research:

Developing a multi-attribute utility function to capture other factors, such as affordability and survival probability, is another research line for future studies in the FCA approach. As mentioned, the current research assumes that the main factor determining utility is graft-survival probability. Although this factor is the most effective, other factors, such as affordability, the recipient’s health condition and PRA and the expected time for recipients to reach the top of the waiting, list may affect the recipient’s utility.

Implementation of the FCA approach in the apportionment of compensation, assessing the potential increase in the number of kidney-exchange chains and the satisfaction level of participating incompatible pairs in these chains.

In summary, we demonstrated that by utilizing the equitable approach, there is more fairness and equity in the allocation of resources in organ-procurement systems, which results in more satisfaction among incompatible pairs. Additional future prospective studies are needed to assess this proposed equitable approach for kidney-exchange chains in countries outside the U.S., such as Iran, that currently use an equality approach.

## Figures and Tables

**Figure 1 jpm-11-01383-f001:**
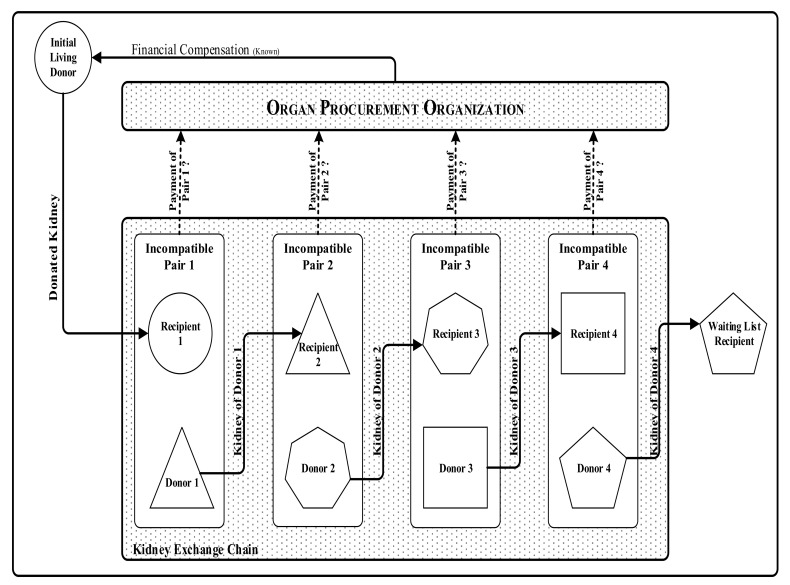
An illustrative statement of the problem under investigation. A kidney-exchange chain with one initiating living donor, four incompatible pairs and a waiting-list recipient. Each incompatible pair makes a payment to the organ-procurement organization, and the total compensation is paid to the initiating donor.

**Figure 2 jpm-11-01383-f002:**
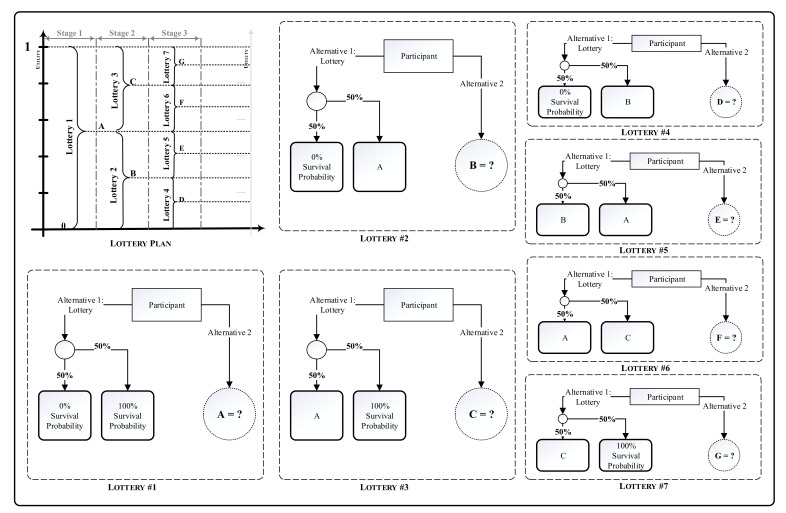
Interview structure based on SG technique. SG, standard gamble.

**Figure 3 jpm-11-01383-f003:**
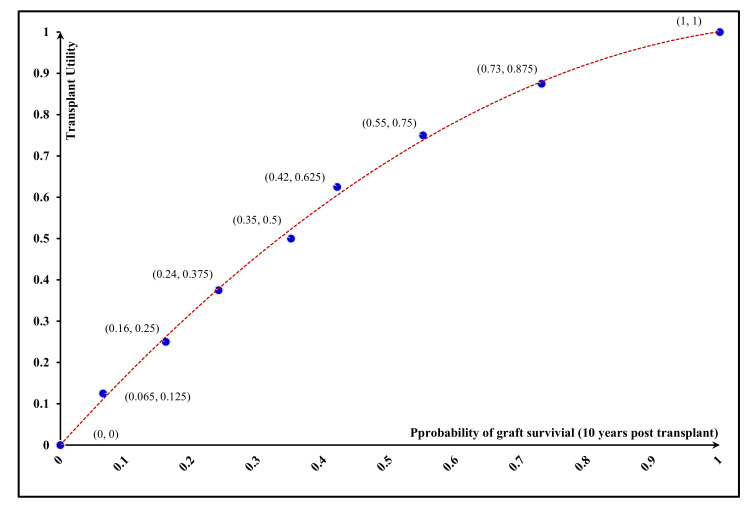
Relationship between probability of graft survival and utility of transplant.

**Figure 4 jpm-11-01383-f004:**
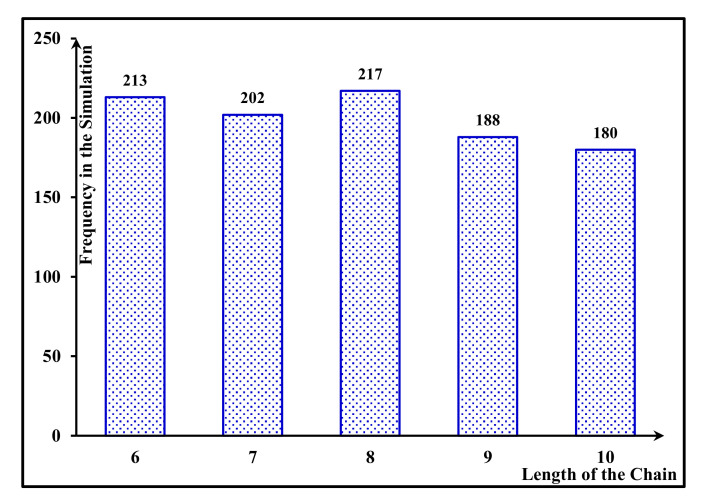
Frequency distribution of chain lengths.

**Figure 5 jpm-11-01383-f005:**
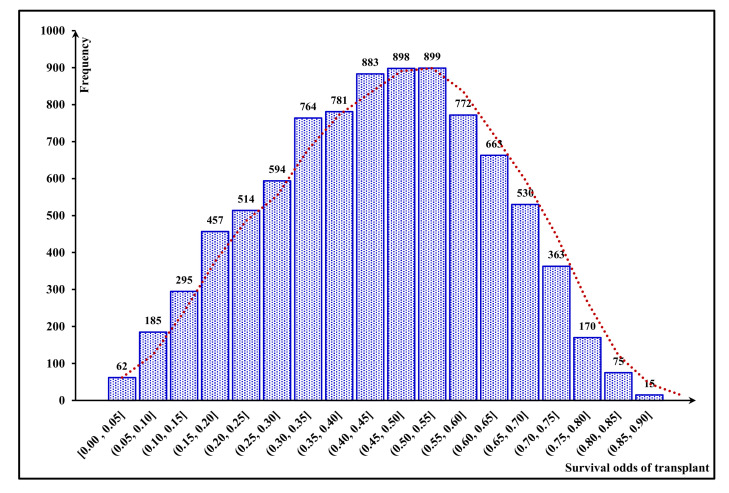
Frequency distribution of 10-year graft-survival probability.

**Figure 6 jpm-11-01383-f006:**
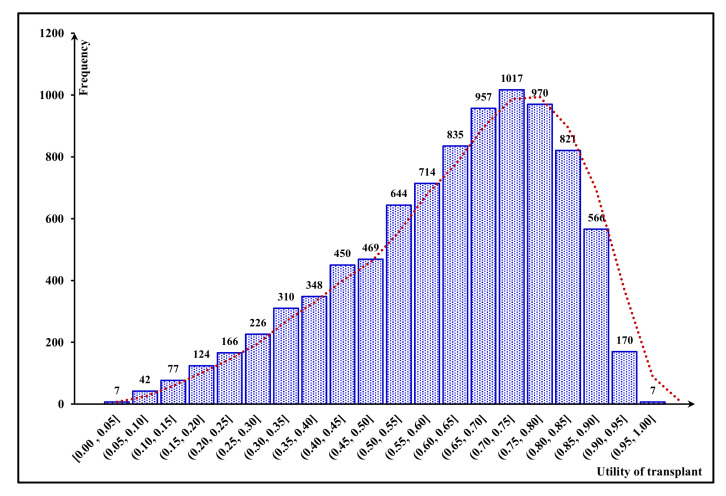
Frequency distribution of utility values.

**Table 1 jpm-11-01383-t001:** Average results of the interviews based on SG technique. SG, standard gamble.

Stage	1	2		3			
Lottery	1	2	3	4	5	6	7
10-Year Survival Probability	35.0%	16.0%	55.0%	6.5%	24.0%	42.0%	73.0%
Utility	0.500	0.250	0.750	0.125	0.375	0.625	0.875

**Table 2 jpm-11-01383-t002:** Characteristics of transplants in the chain in scenarios 1 & 2. HLA, human leukocyte antigen. MM, mismatch. BMI, body mass index. PRA, panel reactive antibody.

Characteristics	Transplant 1 Scenario 1	Transplant 1 Scenario 2	Transplant 2	Transplant 3	Transplant 4	Transplant 5
Living-donor Type	Unrelated	Unrelated	Unrelated	Unrelated	Unrelated	Unrelated
Recipient Age	18–29	18–29	18–29	30–39	18–29	50–59
Donor Age	50–59	<30	30–39	40–49	<30	<30
Donor-Recipient Type	F-M	M-M	M-M	M-F	F-F	M-M
HLA ABDR MM	1–2 HLA, Any DR	1–2 HLA, Any DR	1–2 HLA, 0 DR	0 HLA	0 HLA	1–2 HLA, 0 DR
Recipient BMI	Not Obese (<30)	Not Obese (<30)	Not Obese (<30)	Not Obese (<30)	Obese (>30)	Not Obese (<30)
Donor BMI	Obese (>30)	Not Obese (<30)	Not Obese (<30)	Not Obese (<30)	Not Obese (<30)	Not Obese (<30)
Donor-Recipient Weight Ratio	>1.15	0.90–1.15	0.90–1.15	0.90–1.15	0.90–1.15	0.75–0.90
Donor-Recipient Height Ratio	1.00–1.06	1.00–1.06	1.00–1.06	1.00–1.06	1.00–1.06	0.94–1.00
Recipient Race	White	White	White	White	White	White
Donor Race	Not Black or Hispanic	Not Black or Hispanic	Not Black or Hispanic	Not Black or Hispanic	Not Black or Hispanic	Not Black or Hispanic
Donor History of Cigarette Use	No	No	No	No	Yes	No
ABO Compatibility	Not Incompatible	Not Incompatible	Not Incompatible	Not Incompatible	Not Incompatible	Not Incompatible
PRA	0–9	0–9	10–79	0–9	0–9	0–9
Recipient Diagnosis	Not Diabetes	Not Diabetes	Not Diabetes	Not Diabetes	Not Diabetes	Not Diabetes
Previous Transplant	No	No	No	No	No	No
Time on Dialysis	1–2 Years	1–2 Years	0–1 Years	3+ Years	0–1 Years	2–3 Years
Recipient Hepatitis C Serology	Negative or Missing	Negative or Missing	Negative or Missing	Negative or Missing	Negative or Missing	Negative or Missing
Recipient Insurance	Public Primary Payer	Public Primary Payer	Public Primary Payer	Public Primary Payer	Public Primary Payer	Public Primary Payer
Transplant Year	2008–2012	2008–2012	2008–2012	2008–2012	2008–2012	2008–2012

**Table 3 jpm-11-01383-t003:** Survival probability and utility of transplants in the chain in scenarios 1 and 2.

Survival Pr. and Utility of Transplants		Transplant 1	Transplant 2	Transplant 3	Transplant 4	Transplant 5
Scenario 1	10-Year Survival Probability	46.20%	55.40%	73.70%	57.40%	64.80%
	Estimated Utility	0.65	0.73	0.87	0.75	0.81
Scenario 2	10-Year Survival Probability	52.50%	55.40%	73.70%	57.40%	64.80%
	Estimated Utility	0.71	0.73	0.87	0.75	0.81

**Table 4 jpm-11-01383-t004:** Results of cost-apportionment process in scenario 1.

Total Compensation = $5000	Incompatible Pair 1	Incompatible Pair 2	Incompatible Pair 3	Incompatible Pair 4
Net Utility	−0.09	−0.14	0.12	−0.06
Normalized Net Utility	0.05	0	0.26	0.08
Portion of Total Compensation Assigned	13.60%	0.00%	66.30%	20.10%
Assigned Cost ($)	680	-	3313	1007

**Table 5 jpm-11-01383-t005:** Results of cost-apportionment process in scenario 2.

Total Compensation = $5500	Incompatible Pair 1	Incompatible Pair 2	Incompatible Pair 3	Incompatible Pair 4
Net Utility	−0.03	−0.14	0.12	−0.06
Normalized Net Utility	0.11	0	0.26	0.08
Portion of Total Compensation Assigned	24.90%	0.00%	57.60%	17.50%
Assigned Cost ($)	1368	-	3169	963

**Table 6 jpm-11-01383-t006:** Frequency distribution of 10-year survival outcomes of transplants.

Transplant’s Outcome	Survived	Not-Survived
Frequency	3948	4972
Percentage Frequency	44.3%	55.7%

**Table 7 jpm-11-01383-t007:** Results of cost-apportionment process.

Receiving Transplant’s Outcome	Survived		Not Survived	
Approach	FCA	ECA	FCA	ECA
% of Total Compensation Paid	52%	44%	48%	56%
Average Compensation Paid ($)	740	629	545	633
Participants without Payment	233	-	767	-
% of Incompatible Pairs	44%		56%	

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
