# Peer review of "Equity or Equality? Which Approach Brings More Satisfaction in a Kidney-Exchange Chain?"

_jpm, 2021, doi:10.3390/jpm11121383_

Round 1
Reviewer 1 Report
Dear Authors,
from my humble opinion, this is a very interesting study. Depending the costs of living kidney donation based on survival probability is ethically questionable but financially reasonable. From one point of view, the survival probability is just an estimation, but from the other point, regarding pairs who entered the program seems to be fair.
I read that manuscript with great interest, and I would vote for publication though my physician’s related ethical thoughts still bear in mind.
Kind regards
Tomasz Urbanowicz
Author Response
Response to Reviewer#1
Dear Authors,
from my humble opinion, this is a very interesting study. Depending the costs of living kidney donation based on survival probability is ethically questionable but financially reasonable. From one point of view, the survival probability is just an estimation, but from the other point, regarding pairs who entered the program seems to be fair.
I read that manuscript with great interest, and I would vote for publication though my physician’s related ethical thoughts still bear in mind.
Response: Thank you for reviewing our manuscripts and your critical evaluation. We appreciate your assessment and find our manuscript important.

Reviewer 2 Report
Congratulations for your research, that compares two different approaches based on equality and equity, for a better the share of the compensation for incompatible pairs in kidney transplantation.
The paper is very comprehensive and detailed, bibliography is extensive, but sometimes cited resources are including the same textbooks, but different, still very old editions (1947 and 1953, for example).
It would be a good idea to restructure the manuscript to become more easy to read and follow. I suggest for you to restrict Introduction and General Data, as Methodology starts at page 7. Methodology, a little bit to bushy and hard to follo, but Numerical study is easily readible.
Conclusions could be a little bit more synthetic. Future recommendations are clear and appropriate.
Author Response
Response to Reviewer#2
Congratulations for your research, that compares two different approaches based on equality and equity, for a better the share of the compensation for incompatible pairs in kidney transplantation.
Response: Thank you for reviewing our manuscripts and your critical evaluation.
Comment#1
The paper is very comprehensive and detailed, bibliography is extensive, but sometimes cited resources are including the same textbooks, but different, still very old editions (1947 and 1953, for example).
Response: Thank you for this observation. The repeated citation for " Theory of games and economic behavior" was removed from the manuscript. Another unnecessary citation was also removed from the text according to your comment.
Comment#2
It would be a good idea to restructure the manuscript to become more easy to read and follow. I suggest for you to restrict Introduction and General Data, as Methodology starts at page 7. Methodology, a little bit to bushy and hard to follo, but Numerical study is easily readible.
Response: Thank you for your feedback. In response to your comments, we have thoroughly reviewed and revised sections 1 to 3.1 with the goal of making it more concise and focused on the ultimate goal of this research.
Comment#3
Conclusions could be a little bit more synthetic. Future recommendations are clear and appropriate.
Response: Thank you for your suggestion. As per your feedback, we made modifications to the conclusions section of the manuscript.
Thank you for your time and consideration. We greatly appreciated the reviewer’s and editor’s time and comments to improve our manuscript. The manuscript has been improved considerably by the suggested revisions.
